# HER2-Targeted Immunotherapy and Combined Protocols Showed Promising Antiproliferative Effects in Feline Mammary Carcinoma Cell-Based Models

**DOI:** 10.3390/cancers13092007

**Published:** 2021-04-21

**Authors:** Andreia Gameiro, Catarina Nascimento, Jorge Correia, Fernando Ferreira

**Affiliations:** CIISA—Centro de Investigação Interdisciplinar em Sanidade Animal, Faculdade de Medicina Veterinária, Universidade de Lisboa, Avenida da Universidade Técnica, 1300-477 Lisboa, Portugal; agameiro@fmv.ulisboa.pt (A.G.); catnasc@fmv.ulisboa.pt (C.N.); jcorreia@fmv.ulisboa.pt (J.C.)

**Keywords:** feline mammary carcinoma, HER2, monoclonal antibodies, combined therapies, tyrosine kinase inhibitors, feline *her2* mutations

## Abstract

**Simple Summary:**

Mammary tumors are common in cats, presenting an aggressive behavior with high tumor recurrence. Therefore, new and efficient therapeutic protocols are urgent. Monoclonal antibodies (mAbs; ADC) are widely used in human breast cancer therapy, inhibiting the HER2 dimerization and leading to cell apoptosis. Furthermore, drug combinations, with tyrosine kinase inhibitors (TKi) are valuable in patients’ therapeutic protocols. In this study, two mAbs, and an ADC, as well as combined protocols between mAbs and mAbs plus lapatinib (TKi) were tested to address if the drugs could be used as new therapeutic options in feline mammary tumors. All the compounds and the combined treatments revealed valuable antiproliferative effects, and a conserved cell death mechanism, by apoptosis, in the feline cell lines, where the mutations found in the extracellular domain of the HER2 suggest no immunotherapy resistance.

**Abstract:**

Feline mammary carcinoma (FMC) is a highly prevalent tumor, showing aggressive clinicopathological features, with HER2-positive being the most frequent subtype. While, in human breast cancer, the use of anti-HER2 monoclonal antibodies (mAbs) is common, acting by blocking the extracellular domain (ECD) of the HER2 protein and by inducing cell apoptosis, scarce information is available on use these immunoagents in FMC. Thus, the antiproliferative effects of two mAbs (trastuzumab and pertuzumab), of an antibody–drug conjugate compound (T-DM1) and of combined treatments with a tyrosine kinase inhibitor (lapatinib) were evaluated on three FMC cell lines (CAT-MT, FMCm and FMCp). In parallel, the DNA sequence of the *her2* ECD (subdomains II and IV) was analyzed in 40 clinical samples of FMC, in order to identify mutations, which can lead to antibody resistance or be used as prognostic biomarkers. Results obtained revealed a strong antiproliferative effect in all feline cell lines, and a synergistic response was observed when combined therapies were performed. Additionally, the mutations found were not described as inducing resistance to therapy in breast cancer patients. Altogether, our results suggested that anti-HER2 mAbs could become useful in the treatment of FMC, particularly, if combined with lapatinib, since drug-resistance seems to be rare.

## 1. Introduction

Similarly to human breast cancer [1], the feline mammary carcinoma (FMC) is a very common tumor [2], presenting different molecular subtypes [3], being the feline HER2-positive, the most prevalent one (33–60%) [2,4]. The HER2-overexpression occurs associated to an AKT activation [5], both markers of a poor prognosis and a high metastatic potential [2,6]. In cats, a lack of therapeutic options lead most of the time to surgery [7], the development of different therapeutic strategies being urgent in order to improve the clinical outcome.

The epidermal growth factor receptor 2 (HER2) is a common target in patients with HER2-overexpression tumors [8]. HER2 is a transmembrane glycoprotein, which modulates cell proliferation, differentiation and survival [9,10]. This protein is composed by three domains: an extracellular domain (ECD), a short transmembrane region and an intracellular tyrosine kinase (TK) activity domain [9]. Considering the ECD, it comprises four subdomains, I and II that are the membrane distal regions, and III and IV that are the membrane proximal regions, allowing protein–protein interaction and stabilization [9,11]. Since a sequence identity of 92% was reported between the human and feline *her2* [12,13], the use of monoclonal antibodies (mAbs) in the feline mammary carcinoma could be an alternative and attractive therapeutic approach.

Nowadays, the use of mAbs are widely common in HER2-positive breast cancer patients, decreasing its downstream pathways activation, such as AKT and mTOR, responsible for the antiapoptotic mechanisms [5], cell cycle progression and cell proliferation [14], important checkpoints in the carcinogenesis process [5]. For example, trastuzumab [15,16,17] is a monoclonal IgG antibody that inhibits HER2 homodimerization [18], blocking the HER2 pathway. This antibody prevents the receptor internalization and its degradation, stimulating antibody-dependent cellular cytotoxicity (ADCC) responses [19] and promoting cell apoptosis [20,21]. Trastuzumab resistance is documented in 40% of metastatic patients [22], due to changes in the HER2 expression status [23] and structure [24,25], or other EGFRs pathway’s activators [26,27,28]. In parallel, another mAb commonly used is pertuzumab, a recombinant IgG1 antibody anti-HER2 [15,29], valuable in combined therapeutic protocols [30]. This mAb inhibits the heterodimerization HER2-HER3 [18], inactivating the PI3K pathway and its downstream signaling cascades [31]. This compound also stimulates ADCC [32], leading to cell-cycle arrest and apoptosis [33]. However, specific mutations can lead to pertuzumab resistance [34,35]. More recently, another compound in clinical use for breast cancer treatment is trastuzumab-emtansine (T-DM1) [36], which is an antibody–drug conjugate (ADC), consisting of the humanized monoclonal trastuzumab antibody covalently linked to the cytotoxic tubulin-binding agent (DM1) [37]. T-DM1 mechanism is associated to trastuzumab-HER2 conjugation and the release of the DM1 molecule after complex degradation into lysosomes [38]. The efficacy of this drug depends on the cell membrane HER2 concentration [39] and allows to decrease the systemic cytotoxic effects of DM1, by its specific delivery to the tumor HER2-overexpression cells [38,39], triggering autophagy and apoptosis [40]. T-DM1 leads to the inhibition of HER2 ECD shedding and PI3K/AKT pathway, stimulates ADCC, mitotic arrest, and disruption of the intracellular trafficking. Additionally, for this compound, several different mechanisms of resistance are described, not only associated with the HER2 protein, as well as STAT3 activation [41], defective cyclin B1 induction [42] and multidrug resistance proteins [39].

In human breast cancer, 50% of the HER2-positive patients show immunotherapy resistance [43], the combined therapeutic protocols being truly valuable [44]. Frequently used trastuzumab plus pertuzumab [18,28,44,45] leads to a synergistic response and increased ADCC effects [31,46]. Another common combination is the use of mAbs with tyrosine kinase inhibitors (TKi), which are small molecules that bind to the cytoplasmic catalytic kinase domain of the HER2, preventing tyrosine phosphorylation and signaling [43], such as lapatinib [47,48,49].

An early diagnosis and individual therapy is crucial for the improvement of survival time [50] and prevention of therapy resistance in cats with mammary carcinoma [43]. Thus, this study aims to: (1) evaluate the antiproliferative effects of the mAbs (trastuzumab and pertuzumab) and of the T-DM1 in three feline carcinoma cell lines (CAT-MT, FMCm and FMCp); (2) characterize the HER2 expression in the feline cell lines and identify the existence of genomic mutations in the feline *her2* ECD (subdomains II and IV); (3) describe the cell death mechanism induced by the mAbs and T-DM1, in the carcinoma cell lines; (4) evaluate the synergistic antiproliferative effects by the combination of mAbs (trastuzumab plus pertuzumab) and assess the increase in the cytotoxic response by the use of mAbs with the TKi (lapatinib); (5) identify genomic mutations in the feline *her2* gene ECD (subdomains II and IV) in FMC clinical samples, in order to recognize possible therapy resistant animals, or prognostic factors.

## 2. Materials and Methods

### 2.1. Feline and Human Mammary Carcinoma Cell Lines

In this study, three feline cell lines (CAT-MT from the European Collection of Authenticated Cell Culture, England; FMCp and FMCm kindly provided by Prof. Nobuo Sasaki and Prof. Takayuki Nakagawa, University of Tokyo, Japan) [5,51] and a positive control, the HER2-overexpressing human breast cancer cell line (SKBR-3 from the American Type Culture Collection, Manassas, VA, USA) were used, after characterized by immunocytochemistry (Appendix A and Appendix A). Cell cultures were maintained at 37 °C, in a humidified atmosphere of 5% (*v*/*v*) CO_2_ (Nuaire, Plymouth, MN, USA), in Dulbecco’s Modified Eagle Medium (DMEM; Corning, New York, NY, USA), for CAT-MT and SKBR-3, whereas FMCm and FMCp cell lines were maintained in Roswell Park Memorial Institute 1640 Medium (RPMI; Corning), both supplemented with heat-inactivated 20% (*v*/*v*) fetal bovine serum (FBS; Corning). Periodically, all cell lines were inspected to control their morphology and proliferation rate, being tested for Mycoplasma (MycoSEQ™ Mycoplasma Detection Kit, Thermo Fischer Scientific, Waltham, MA, USA).

### 2.2. In Vitro Cytotoxicity Assays

Viability assays were performed to determine the antiproliferative effects of trastuzumab, pertuzumab and T-DM1 (all from Roche, Basel, Switzerland), using the Cell Proliferation Reagent WST-1 (Abcam, Cambridge, United Kingdom) and following the manufacturer’s instructions. Briefly, cell lines were seeded in 96-well plates to obtain a confluency of 90%, after 24 h (5 × 10^3^ cells/well for CAT-MT and FMCp, 15 × 10^3^ cells/well for FMCm and 10 × 10^3^ cells/wells for SKBR-3), and then exposed to increasing concentrations of each antibody (Table 1), with the control wells left unexposed. Phosphate buffered saline (PBS; Corning) was used as a vehicle for mAbs and ADC. After 72 h of exposure, the WST-1 reagent (Abcam) was added, followed by an incubation period of 4 h, at 37 °C, and absorbance was measured at 440 nm using a plate reader (FLUOStar Optima, BMG LabTech, GmbH, Ortenberg, Germany). Triplicate wells were used to determine each data point and three independent experiments were performed.

For the combined assays: trastuzumab plus pertuzumab, trastuzumab plus lapatinib (Sigma-Aldrich, Darmstadt, Germany) and pertuzumab plus lapatinib, a similar methodology was used, testing concentrations that covers different cytotoxic responses (Table 2).

### 2.3. Assessment of HER2 Expression Status by Immunocytochemistry (ICC)

Expression analysis of HER2 was performed, as reported by us [4,52], in the three feline cell lines (CAT-MT, FMCm and FMCp), and using the human SKBR-3 cell line as a positive control. Briefly, cells were grown till confluency in a T25 culture flask and then were removed and embedded in a histogel matrix (Thermo Fischer Scientific). Cytoblocks were sectioned in slices with 3 µm thickness (Microtome Leica RM135, Newcastle, UK) and mounted on a glass slide (SuperFrost Plus, Thermo Fisher Scientific). On PT-Link module (Dako, Agilent, Santa Clara, CA, USA) samples were deparaffinized, hydrated and antigen retrieval was performed for 20 min at 96 °C, using citrate buffer pH 6.1 (EnVision™ Flex Target Retrieval Solution Low pH, Dako). Then, slides were cooled for 30 min at room temperature (RT) and immersed twice in distilled water for 5 min. ICC technique was performed with commercial solutions from the EnVision™ FLEX+, Mouse kit (Dako). Before antibody incubation, samples were treated with Peroxidase Block Novocastra Solution (Leica Biosystems) for 15 min. Afterwards, samples were incubated with a primary antibody anti-HER2 (clone CB11, 1:100, ab8054; Abcam) by 1 h at RT, in a humidified chamber. Then, the EnVision™ FLEX+ Mouse Linker was incubated by 30 min and slides were washed for 5 min, between all the incubation steps, using PBS at pH 7.4. Later on, the EnVision™ FLEX/HRP was incubated for 30 min at RT, and detection was performed using diaminobenzidine (DAB substrate buffer and DAB Chromogen, Dako) for 5 min. Finally, samples were counterstained with Gill’s hematoxylin (Merck, Darmstadt, Germany) for 5 min, dehydrated in an ethanol gradient and xylene, and mounted using Entellan mounting medium (Merck).

HER2 immunoreactivity was scored as recommended by the American Society of Clinical Oncology’s (ASCO) [53], and as previously published for feline cells [2,4]. Briefly, the staining intensity was evaluated and classified as HER2-negative when scored 0 and HER2-positive if scored 1+, 2+ or 3+ (Table 3). Three microscopic fields were analyzed at 400× magnification. All samples were subjected to a blind scoring, by two independent pathologists.

### 2.4. Flow Cytometry Assay

The flow cytometry assay was performed as already reported by us [52]. For this experiment, the cells were seeded in 24-well plates to obtain a confluency of 90% after 24 h (5 × 10^4^ cells/well for CAT-MT and FMCp, 10 × 10^4^ cells/well for FMCm, and 7 × 10^4^ cells/well for SKBR-3 cell lines), and then were exposed to mAbs and to the ADC for 72 h, at a concentration close to the EC_50_ value. Control wells were left unexposed, with the PBS used as a vehicle, in three independent experiments. The percentage of apoptotic cells after drug exposure was calculated by using the APC Annexin V Apoptosis Detection Kit with Propidium Iodide (PI; BioLegend, San Diego, CA, USA) and following the manufacturer’s instructions. Briefly, supernatants were harvested, and the remaining attached cells were trypsinized (Trypsin-EDTA; Corning) and added to the correspondent supernatants. Then, samples were centrifuged for 5 min at 500 g at RT, washed with PBS and resuspended in 500 µL of Annexin V Binding Buffer (BioLegend), with a maximum concentration of 1 × 10^7^ cells/mL. Afterwards, 100 µL of each sample were passed to a new tube, and 5 µL of APC Annexin V (BioLegend) and 10 µL of PI (BioLegend) was added. Then, samples were vortexed and incubated for 15 min at RT, protected from the light. Finally, 400 µL of Annexin V Binding Buffer (BioLegend) were added to the samples, before acquisition in a BD LSR Fortessa X-20 (BD Biosciences, San Jose, CA, USA), at Champalimaud Foundation, Lisbon, Portugal. Data were analyzed using FlowJo software (v.10.7.1, for Windows, BD Biosciences), considering double negative staining cells, as living cells, annexin positive cells in early apoptosis and double positive cells (both for annexin and PI) in late apoptosis phase.

### 2.5. Animal Population

The 40 tumor tissue samples used in this study were collected from cats that underwent mastectomy, at the Teaching Hospital of the Faculty of Veterinary Medicine, University of Lisbon, with all the procedures consented by the owners, showing no interference in the animals’ well-being. The clinical history of the cats was recorded (Table 4), including breed, age, reproductive status and contraceptive administration, treatment (mastectomy or mastectomy plus chemotherapy), number, location and size of tumor lesions, histopathological classification, ER status, PR status, HER2 status and Ki-67 index, malignancy grade, tumor necrosis, lymphatic invasion, lymphocytic infiltration, cutaneous ulceration, regional lymph node involvement, and clinical stage (TNM system) [2], with tumors being classified into five molecular subtypes [3]. Additionally, the fHER2 was determined as being overexpressed in 33% of the FMC cases, although *her2* gene amplification was not observed [4]. All samples were frozen at −80 °C and stored until further use.

### 2.6. DNA Extraction, Amplification and Sequence Analysis of Feline HER2 ECD

Genomic DNA extraction was performed in 5 mg of 44 frozen tissue samples (4 breed control and 40 tumor tissues) and in the three feline cell lines (CAT-MT, FMCm and FMCp), collected after grown till confluency in a T25 culture flask, as previously described [54,55], using a QIAmp FFPE kit (Qiagen, Dusseldorf, Germany) and following the manufacturer’s guidelines. First, tissue samples were homogenized in the Tissue Lyser II (Qiagen), and then, all samples were digested with protease K (20 mg/mL; Qiagen). After the washing steps, genomic DNA was eluted from the extraction columns and its quality and quantity was measured in NanoDrop ND-100 Spectrophotometer (Thermo Fischer Scientific). For the amplification of exons 3, 4 and 10–13, which encode for the feline *her2* ECD (NC_018736.3), previously identified by comparison with the genomic human *her2* sequence (NC_000017.11), different primers were designed (Table 5) in the Primer designing tool (NCBI, Bethesda, MD, USA), and used in a PCR Thermal Cycler (VWR Thermocycler, Leicestershire, England). PCR technique was performed with a standard reaction mixture (4 µL/sample of Phusion GC buffer (Thermo Fischer Scientific), 0.4 µL/sample of dNTPs (Grisp, Porto, Portugal), 0.1 µL/sample of each forward and reverse primers and 0.2 µL/sample of DNA Polymerase (Thermo Fischer Scientific)), maintaining a final DNA concentration of 4 ng/mL. For exons 3, 4, 10 and 11, PCR reactions were performed as follows: denaturation at 98 °C for 30 s, followed by 35 cycles at 98 °C for 10 s, 58 °C for 30 s, 72 °C for 10 s, plus one final extension step at 72 °C for 10 min. For exons 12 and 13 a nested-PCR was performed, using two pairs of primers, initially with a melting temperature of 55 °C, followed by a melting temperature of 52 °C for the second pair of primers. After confirmation of the expected size for each amplified sequence in a 2% agarose gel (Sigma-Aldrich), DNA fragments were purified and sequenced by Sanger technique (StabVida, Almada, Portugal), and checked for inaccuracies.

The feline sequenced samples were aligned with the identified feline *her2* (NC_018736.3), using the ClustalW tool (BioEdit Alignment Editor software) [56], while the consensus sequence was confirmed using the SeqTrace 9.1 software [57]. Protein mutations and SNP loci were identified using the Expert Protein Analysis System (ExPASY) translate tool and compared with the original protein sequence (NP_001041628.1, NCBI). Mutations identified in the feline tissue samples were compared to the human *her2* sequence (NC_000017.11) and searched in National Cancer Institute, International Cancer Genome Consortium and Catalogue of Somatic Mutations in Cancer (COSMIC) databases for putatively induced resistance to mAbs, and the ADC tested in this study.

### 2.7. Statistical Analysis

Statistical analysis was carried out using the GraphPad Prism software (version 5.04, for Windows, San Diego, CA, USA), with two-tailed *p*-values below 0.05 being considered statistically significant, and a 95% confidence interval (* *p* < 0.05, ** *p* < 0.01 and *** *p* < 0.001). For the cytotoxicity assays, outliers with more than two standard deviations were removed from the analysis and the EC_50_ value for each drug, in all the cell lines was calculated using Log (agonist) vs. Response (Variable slope) function. In the drug conjugation assays, the two-way ANOVA test was performed. Regarding the mutations found in tissue samples, associations between groups were assessed using the nonparametric Fisher’s exact test.

## 3. Results

### 3.1. Trastuzumab, Pertuzumab and T-DM1 Presented Antiproliferative Effects in FMC Cell Lines

Trastuzumab, pertuzumab and T-DM1 exhibited a dose-dependent antiproliferative effect on feline mammary carcinoma cell lines. Indeed, trastuzumab exerted a more potent antiproliferative effect on CAT-MT cell line (92.6% of cytotoxicity, EC_50_ = 3047.89 µg/mL ± 1.43; Figure 1A) in comparison with the effects on the FMCm (82.7% of cytotoxicity, EC_50_ = 528.45 µg/mL ± 1.14) and FMCp (60.1% of cytotoxicity, EC_50_ = 3243.40 µg/mL ± 2.29) cell lines. In parallel, pertuzumab showed similar results on CAT-MT and FMCm cell lines (60.2% and 61.8% of cytotoxicity, EC_50_ = 2837.92 µg/mL ± 1.50 and EC_50_ = 1205.04 µg/mL ± 1.23, respectively; Figure 1B), with the feline HER2-negative cell line, FMCp presenting the lowest antiproliferative effect (52.1% of cytotoxicity, EC_50_ = 928.97 µg/mL ± 1.11). Regarding the ADC T-DM1, the highest antiproliferative effect, at the range of concentrations tested, was reported on CAT-MT and FMCp cells (94.0% and 74.2% of cytotoxicity, EC_50_ = 19.63 µg/mL ± 1.22 and EC_50_ = 88.72 µg/mL ± 1.29, respectively; Figure 1C), with similar cytotoxicity being detected on FMCm and SKBR-3 cell lines (53.8% and 50.5% of cytotoxicity, respectively; EC_50_ (FMCm) = 52.84 µg/mL ± 1.50).

In order to interpret the response of cell lines to anti-HER2 agents, the HER2 expression status was determined. Interestingly, all feline cell lines showed lower HER2 scores than the human SKBR-3 cell line (scored as 3+, Figure 2D), with CAT-MT cells showing a 2+ score (Figure 2A), FMCm cells showing 1+ HER-2 positivity (Figure 2B) and FMCp cells showing no immunostaining signal (0 score, Figure 2C).

### 3.2. Apoptosis Is the Main Mechanism of Cell Death Caused by Anti-HER2 mAbs and ADC T-DM1

Flow cytometry analysis of feline cell lines exposed to anti-HER2 mAbs (trastuzumab and pertuzumab) or ADC (T-DM1) allowed to confirm that the observed antiproliferative effects were related to apoptosis induction (Figure 3). Deepening the analysis, we observed that the HER2-overexpressing CAT-MT cell line showed the higher percentage of cells in late apoptosis for both mAbs, reaching a maximum of 53.1% ± 2.54 of apoptotic cells after trastuzumab exposure (14.1% early apoptosis and 39.0% late apoptosis) and 65.3% ± 2.33 after treated with pertuzumab (12.8% early apoptosis and 52.5% late apoptosis; Figure 3A). The balance between early and late apoptosis was similar when cells were treated with T-DM1 (41.7% and 39.6%, respectively), with a maximum of 81.3% ± 1.61 of apoptotic cells. In FMCm cell line the percentage of cells in late apoptosis was the highest, presenting 52% ± 4.02 of apoptotic cells (10.5% early apoptosis and 41.5% late apoptosis) after exposure to trastuzumab, 44.4% ± 1.04 as response to pertuzumab (12.1% early apoptosis and 32.3% late apoptosis) and 74.2% ± 6.06 of apoptotic cells (25.1% early apoptosis and 49.1% late apoptosis) after T-DM1 incubation (Figure 3B). A different behavior was observed in the FMCp cell line that showed a higher percentage of cells in early apoptosis. The total percentage of apoptotic cells after exposure to trastuzumab was 44.5% ± 1.64 (36.4% early apoptosis and 8.1% late apoptosis), and 52.3% ± 1.79 (47.1% early apoptosis and 5.2% late apoptosis) after pertuzumab exposure. In these cells, T-DM1 induced 44.5% ± 2.63 of cellular apoptosis (41.9% early apoptosis and 2.6% late apoptosis; Figure 3C). Lastly, the human SKBR-3 cells used as control, showed expected results, with an average of 48.8% ± 1.62 of apoptotic cells when exposed to trastuzumab and pertuzumab, similarly distributed between early and late apoptosis stages (average 22.2% and 26.6%, respectively), with the percentage of apoptotic cells being higher after T-DM1 exposure (80.1% ± 2.00; 27.5% early apoptosis and 52.6% late apoptosis; Figure 3D).

### 3.3. Combined Exposures of Two Anti-HER2 mAbs and Anti-HER2 mAbs with Lapatinib Showed Synergistic Antiproliferative Effects

The combination trastuzumab plus pertuzumab showed synergistic antiproliferative effects in all feline cell lines. In CAT-MT cells the exposure to trastuzumab at 125 µg/mL plus pertuzumab at 2000 µg/mL increased the cytotoxicity from 13.6% to 40.0% (*p* < 0.001; Figure 4A). For the FMCm cells, the cytotoxic effect was increased from 15.6% to 45.1% (*p* < 0.001; Figure 4B), after exposure to trastuzumab at 125 µg/mL plus pertuzumab at 2000 µg/mL. Finally, in the FMCp cell line, the highest antiproliferative effect was obtained using trastuzumab at 500 µg/mL plus pertuzumab at 2000 µg/mL, increasing the cytotoxicity from 17.1% to 28.8% (*p* = 0.018; Figure 4C). In the human SKBR-3 cells, all the conjugations tested exerted a significant increase of the antiproliferative effect (*p* < 0.001; Figure 4D).

The use of the anti-HER2 mAbs (trastuzumab and pertuzumab) combined with lapatinib showed an improvement of the antiproliferative effects induced by mAbs. Accordingly, the combination of trastuzumab plus lapatinib, in the CAT-MT cell line, revealed a valuable synergistic response, for all the concentrations tested, with the highest increase of 78.4% of cytotoxicity, by the combination of trastuzumab at 125 µg/mL plus lapatinib at 7.26 µg/mL (*p* < 0.001; Figure 5A). The other two feline cell lines (FMCm and FMCp) presented different behaviors. While, the FMCm cells presented a lower response, with the highest antiproliferative effect occurring by exposure to trastuzumab at 125 µg/mL plus lapatinib at 7.26 µg/mL, increasing the cytotoxicity from 10.2% to 37.4% (*p* = 0.0017; Figure 5B), the FMCp cells had a significant decrease in its proliferation rate for all the combinations tested (*p* < 0.001; Figure 5C), with a maximum cytotoxic effect of 84.6% of cytotoxicity, by combining trastuzumab at 500 µg/mL plus lapatinib at 7.26 µg/mL. In the SKBR-3 cell line all the concentrations tested presented an effective synergistic response (*p* < 0.001; Figure 5D).

Similarly to the previous results, the combination of pertuzumab plus lapatinib also revealed to be valuable, increasing the mAb’s antiproliferative effects. In fact, the combination of pertuzumab at 250 µg/mL plus lapatinib at 7.26 µg/mL, in the CAT-MT cells revealed a synergistic effect, with an increase from 7.4% to 76.8% of cytotoxicity (*p* < 0.001; Figure 6A). Again, the FMCm cell line presented to be the less sensitive, a maximum antiproliferative effect being obtained by exposure to pertuzumab at 250 µg/mL plus lapatinib at 7.26 µg/mL, increasing the cytotoxicity from 6.3% to 47.8% (*p* < 0.001; Figure 6B). In the FMCp cells the highest increase in the cytotoxic effect was observed using pertuzumab at 250 µg/mL plus lapatinib at 7.26 µg/mL, increasing the cytotoxicity from 6.2% to 53.7% (*p* < 0.001; Figure 6C). As control, the human cell line SKBR-3 also presented an effective increase in the antiproliferative effects of the combined assays (*p* < 0.001; Figure 6D), for all the concentrations tested.

### 3.4. In the FMC Clinical Samples, Mutations Found in the HER2 ECD, Subdomains II and IV Were Not Associated with Immunotherapy Resistance in Humans

Previous studies on somatic mutations in the *her2* gene, in human breast cancer patients, revealed several mutations associated with resistance to therapy and/or specific clinicopathological features, reported in National Cancer Institute, International Cancer Genome Consortium and COSMIC databases. In this work, the FMC clinical samples revealed that 45% (18/40) presented at least one mutation (Figure 7; Appendix A), with all of them showing a low frequency (n = 1), and the majority being homozygous (83.3%; 30/36 mutations). Detected mutations were more frequent in luminal B tumors (61.1%; 11/18), followed by the luminal B/HER2-positive subtype (50%; 4/8) and triple-negative tumor samples (42.9%; 3/7), with no mutations identified in HER2-positive (n = 4) and luminal A (n = 3) tumor subtypes. Further analysis revealed that in the region recognized by pertuzumab, *her2* subdomain II, while 69.4% (25/36 mutations) of the mutations were located in exon 3 (11/40 tumor samples), no mutations were found in the exon 4. Additionally, exon 3 presented 11 synonymous and 10 missense mutations, the remaining four being silent mutations. Regarding the subdomain IV, which is the *her2* region recognized by trastuzumab, few mutations were detected, namely one heterozygous missense mutation at exon 10 (2.8%; 1/36), in two tumor samples; five mutations in exon 11 (13.9%; 5/36 mutations), one being a heterozygous synonymous mutation and four missense mutations, occurring in three tumor samples; two mutations at exon 12 (5.6%; 2/36 mutations), occurring in two tumor samples, one of them synonymous; three mutations at exon 13 (8.3%; 3/36 mutations), in two tumor samples, one synonymous, one missense and one frameshift mutation (c.14406 Ins. C).

Additionally, any correlation was reported between animals’ clinicopathological features and the described mutations. Moreover, after comparing the feline mutations with the genomic human *her2* ECD sequence, and searching in the referred databases, the mutations found were never reported as inducing resistance to anti-HER2 immunotherapy in breast cancer patients.

## 4. Discussion

The FMC is a common disease, showing similar features with human breast cancer [1,2]. Presently, a lack of therapeutic options combined with a late diagnosis have a severe impact on overall survival and disease-free survival of cats with mammary carcinoma. Thus, to overcome this problem, and since the cat is a good cancer model [58], the antiproliferative effects of two anti-HER2 mAbs (trastuzumab and pertuzumab) and one ADC (T-DM1) were evaluated, using a FMC cell-based model (CAT-MT, FMCm and FMCp cell lines), presenting different HER2 expression levels, and with no mutations described as conducing to resistance to therapy (Appendix A).

Currently, only one felinized mAb is known to be in use as a therapeutic option for cats [59,60], with no information about the use of mAbs in the treatment of the FMC. Furthermore, despite the reproducible [61,62] and promising results obtained by testing human TKi, approved for breast cancer patients (lapatinib and neratinib), in FMC cell-based models, achieving 100% of cytotoxicity by the use of lapatinib (average IC_50_ = 8756 nM ± 83), and a maximum of 79.4% of cytotoxicity in the FMCp cell line, by exposure to neratinib [63], no clinical trials are performed. In cats, the approved TKi are not specific for the mammary tumors, and are associated with severe side effects [64]. Thus, the immunotherapy protocols, using mAbs could arise has an alternative to chemotherapy. Accordingly, trastuzumab avoids HER2 homodimerization and its ECD shedding, leading to HER2 endocytic destruction. Furthermore, this mAb promotes an immune activation [20], inducing apoptosis, by the reduction of the antiapoptotic Mcl-1 protein expression [65], and the inhibition of PI3K/AKT [66], a crucial cell survival pathway [5]. This study demonstrated that trastuzumab presented similar effects between the feline and human cell lines [67,68], being possible to report a maximum of 92.6% of cytotoxicity for the CAT-MT cells and 82.7% of cytotoxicity for the FMCm cells, both HER2-positive. Particularly, for the FMCm cell line it was demonstrated that the HER2 expression levels are an upstream activator of the AKT pathway [5], demonstrating this mAb an important effect, by blocking the HER2 protein and contradicting this cycle. On the other hand, for the HER2-negative FMCp cell line, the cytotoxic effect obtained was the lowest (60.1% of cytotoxicity), suggesting that the lack of HER2 receptor [69] decreases trastuzumab efficacy. Additionally, such as described in other studies, the HER2 expression in the FMCp cell line is not completely null [5,51]. This result could be associated to the expression of an activated HER2 (Y877) protein, already described in triple-negative human tumors, being responsible for the benefits of trastuzumab in some breast tumor patients [70]. Furthermore, similar results between cell lines were obtained by the use of pertuzumab, as reported in other studies [68,71,72]. Pertuzumab inhibits heregulin induced heterodimerization HER2-HER3 [73], with a decrease in the activation of the PI3K [74] and ERK pathways [75]. Analyzing this assay, it was possible to obtain 60.2% and 61.8% of cytotoxicity, for CAT-MT and the metastatic FMCm cell lines, respectively. Interestingly, similarly to the trastuzumab effect, FMCp cells exposed to pertuzumab, revealed a good cytotoxic response (52.1% of cytotoxicity). In fact, pertuzumab was suggested for the treatment of triple-negative human breast cancer, which express the circular HER2 RNA, encoding for HER2-103, as putatively occurring with the FMCp cell line [51], which is antagonized by pertuzumab [76]. Finally, ADC T-DM1 allows a selective delivery of the DM-1 molecule to the HER2-expressing tumor cells, preventing the HER2 homodimerization, and inhibiting the microtubule assembly [77], which induces cell apoptosis [39,40,77], by blocking the AKT/mTOR pathway [40]. In the experimental assay, T-DM1 was tested in lower concentrations, comparing to mAbs, and showing superior cytotoxic effects, directly dependent on the cell membrane HER2 concentration [39]. In fact, 94.0% of cytotoxicity was obtained for the CAT-MT cells, presenting a HER2 2+ score, and 53.8% of cytotoxicity for the metastatic FMCm cells, with a HER2 1+ score. Interestingly, the cytotoxic effect of T-DM1 in the FMCp cell line presented promising results, 74.2% of cytotoxicity being obtained, which could be explained by the axis DM-1/cytoskeleton-associated protein 5 (CKAP5) a microtubule regulator protein, cell surface target for T-DM1, inducing cytotoxicity in no HER2-overexpressing cells [78], suggested as a target in triple-negative breast cancer therapy [79]. Furthermore, we were unsuccessful in reproducing the cytotoxic effect in the SKBR-3 cell line after T-DM1 exposure [42,78,80], obtaining 50.5% of cytotoxicity, at the maximum concentration tested (1000 µg/mL). Although the general characteristics of the tumor are maintained (Appendix A and Appendix A), this result could be explained due to molecular changes along cell culture [81,82]. Unfortunately, it was not possible to achieve 100% of cytotoxicity, with any of the compounds tested, which could be explained due to a need of a 3D system, allowing a proper antigen–antibody conformational interaction [71,83].

The tested mAbs and the ADC cell death mechanism occur by apoptosis, suggesting a Bcl-2 dependent mechanism [84], conserved between the feline and human cell lines [33,39,65]. Furthermore, the majority of the cells occurred in late phase apoptosis after drug exposure, presenting a permeabilized cell membrane [85] (Annexin positive vs. PI positive). Nevertheless, in the FMCp cell line a higher percentage of cells occur in early phase apoptosis (Annexin positive vs. PI negative), for all the compounds tested, suggesting the need of a more prolonged exposure time to achieve the late phase apoptosis.

As for human breast cancer, combined protocols reveal to be a valuable tool [44,86,87] in the feline mammary tumor cell lines, presenting antiproliferative synergistic effects, by the use of smaller drug concentrations, and preventing the acquired resistance to therapy, known for trastuzumab [88,89,90,91,92,93], as well as for pertuzumab [94]. In the conjugation trastuzumab plus pertuzumab the obtained response was known by block heterodimers formation, canceling the influence of HER2-overexpression on cell cycle and blocking the signaling through AKT [45]. This combination reveals to be valuable for all the feline cell lines, with the highest increase in cytotoxicity in the FMCm cell line (from 15.6% to 45.1% of cytotoxicity). Better results were obtained, considering the conjugations of mAbs plus the TKi, lapatinib, already approved for combined protocols in human breast cancer [95,96,97], and revealing valuable in FMC cell-based models [63]. In the combined treatments similar results to the SKBR-3 cell line [47] were obtained, revealing for all the combinations a synergistic behavior. The best conjugation reveals to be trastuzumab plus lapatinib, with a highest increase of 71.9% of cytotoxicity (from 6.52% to 78.4% of cytotoxicity), in the CAT-MT cell line. This combination is valuable since the TKi enhances trastuzumab mediated ADCC, by upregulating the HER2 expression [97]. Nevertheless, the conjugation pertuzumab plus lapatinib also presented a good improvement of the antiproliferative effects, with the highest increase of 69.4% of cytotoxicity (from 7.4% to 76.8% of cytotoxicity), also occurring in the CAT-MT cell line. In this study we associated mAbs with the reversible TKi, lapatinib, but combined protocols of mAbs with the irreversible TKi, neratinib, have been revealing valuable tools for the treatment of breast cancer patients [98]. Furthermore, considering the promising results obtained by the use of neratinb, in the FMC cell lines [63], more studies are needed with neratinib in conjugation assays, to unveil different possibilities for the treatment of cats with mammary carcinomas.

Beyond the acquired therapy resistance, somatic mutations in the *her2* ECD could explain the resistance patterns obtained with the use of anti-HER2 molecules, although mutations in the *her2* gene are described as more common in the TK domain [35], and in triple-negative tumor subtype [99]. In the FMC clinical samples, mutations revealed to occur with higher frequency in the luminal B subtype (61.1%). In the studied population, the majority of the *her2* ECD mutations occurred in exon 3 (69.4%), subdomain II, which encode for the region recognized by pertuzumab, suggesting that if resistance occurs in cat, it might be more common in therapeutic protocols using this mAb. Additionally, 38.9% (14/36 mutations) of the mutations were synonymous and 61.1% (22/36 mutations) corresponds to missense mutations, conditioning 59.1% (13/22 mutations) of them a change in the polarity of the codified codon, which could modify the 3D arrangement of the protein [100]. In breast cancer patients, one of the most described mutations in the *her2* ECD is S310F/Y, related to a resistant pattern to trastuzumab [101] and pertuzumab [35,102] therapies, not reported in our population. Furthermore, none of the mutations found in the *her2* ECD were described as inducing resistance to therapy, according to National Cancer Institute, International Cancer Genome Consortium and COSMIC databases, and none of them were related to the cat’s clinicopathological features.

## 5. Conclusions

This study tested two mAbs (trastuzumab and pertuzumab) and an ADC compound (T-DM1) revealing valuable in vitro antiproliferative effects in feline primary (CAT-MT and FCMp) and metastatic (FMCm) cell lines, as well as a conserved cell death mechanism, by apoptosis, comparing to the human breast cancer cell line, SKBR-3. Furthermore, promising synergistic antiproliferative effects were obtained by combining both mAbs (trastuzumab plus pertuzumab), and mAbs with the TKi, lapatinib, already referred to as a valuable therapeutic protocol for humans. Additionally, mutations found in the clinical samples were never reported as inducing resistance to therapy, in breast cancer patients. Concluding, the use of anti-HER2 mAbs and combined protocols are proposed as a new targeted therapy for cats with different mammary carcinoma subtypes, suggesting that mAbs-resistant FMCs are rare. Still, more studies are needed to produce felinized anti-HER2 monoclonal antibodies, preventing cat immune reactions. Moreover, the similarities between the FMC and human breast cancer were demonstrated, reinforcing the utility of the cat as a breast cancer model.

## Figures and Tables

**Figure 1 cancers-13-02007-f001:**
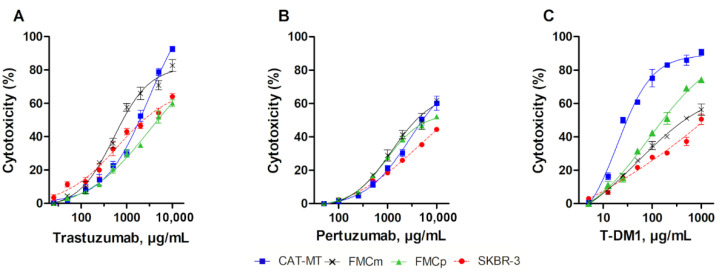
Trastuzumab, pertuzumab and T-DM1 presented strong antiproliferative effects on feline mammary carcinoma cell lines. (**A**) CAT-MT cells exposure to trastuzumab allowed to obtain a maximum of 92.6% of cytotoxicity. (**B**) Cell lines exposure to pertuzumab presented a lower antiproliferative effect, with a maximum of 61.8% of cytotoxicity in the FMCm cells. (**C**) T-DM1 had the highest antiproliferative effect in the CAT-MT cells, with 94.0% of cytotoxicity. The experiments were performed in triplicates, in three independent assays.

**Figure 2 cancers-13-02007-f002:**
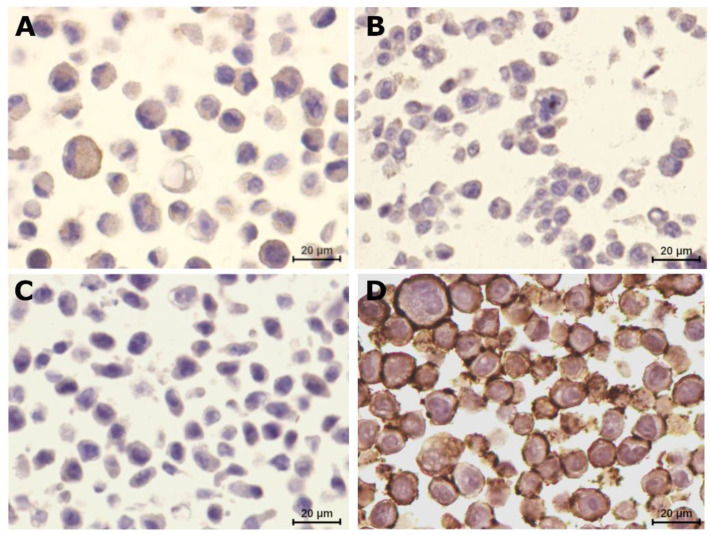
Feline mammary carcinoma cell lines presented different HER2 expressions. (**A**) CAT-MT cell line was classified as HER2-positive (HER2 2+ score). (**B**) FMCm was classified with 1+ score, being considered slightly positive for HER2 expression. (**C**) FMCp cell line presented no HER2 signal (HER2-negative). (**D**) The human SKBR-3 cell line was classified as a HER2-overexpressing cell line (3+ score). 400× magnification.

**Figure 3 cancers-13-02007-f003:**
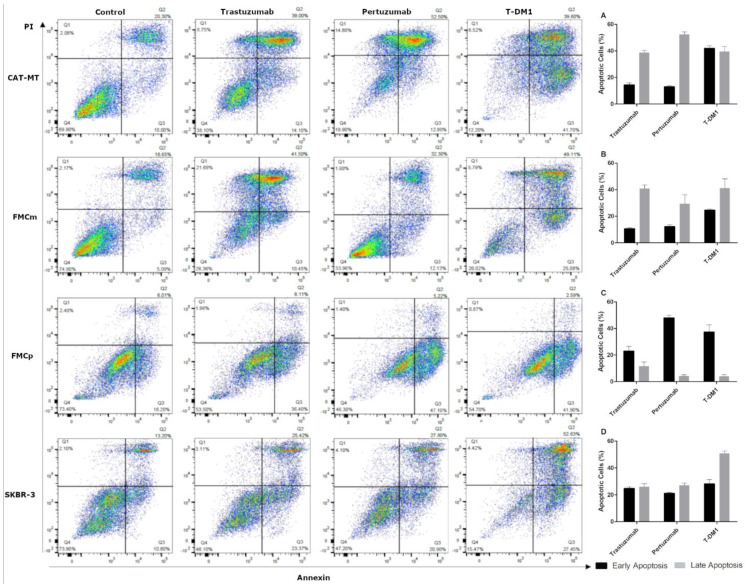
Trastuzumab, pertuzumab and T-DM1 induce apoptosis in the feline cell lines. (**A**) CAT-MT and (**B**) FMCm cell lines presented an increase in cell apoptosis after drugs exposure, being the cells predominantly in late apoptosis phase. (**C**) In the FMCp cell line, cells occurred predominantly in early stage apoptosis. (**D**) The human SKBR-3 cell line had an equilibrated death pattern, between early and late stage apoptosis, with the cells mainly in late apoptosis phase, by exposure to T-DM1. The experiment was performed in triplicates and repeated simultaneously three times.

**Figure 4 cancers-13-02007-f004:**
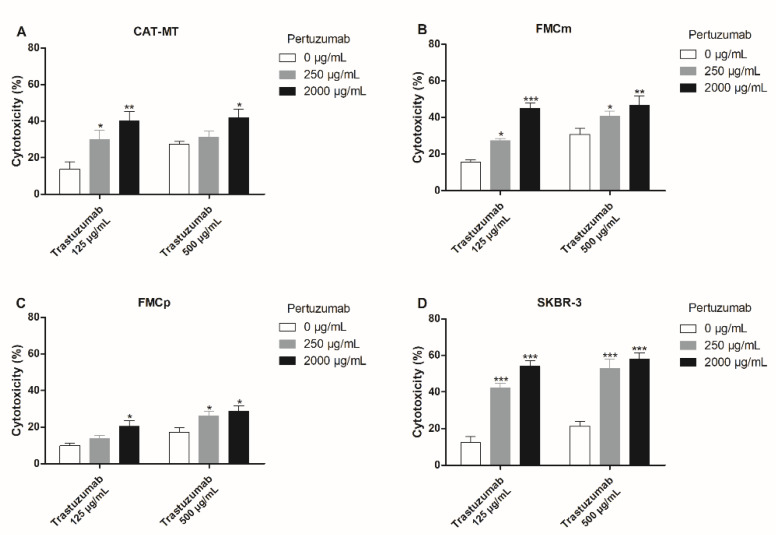
Combined treatments with trastuzumab plus pertuzumab showed synergistic antiproliferative effects in the carcinoma cell lines. (**A**) CAT-MT cells presented the highest cytotoxic response by the use of trastuzumab at 125 µg/mL plus pertuzumab at 2000 µg/mL, being possible to achieve 40.0% of cytotoxicity (** *p* < 0.01). (**B**) In FMCm cell line, the combination trastuzumab at 125 µg/mL plus pertuzumab at 2000 µg/mL was the one that allows to achieve the highest cytotoxic effect, 45.1% of cytotoxicity (*** *p* < 0.001). (**C**) In the FMCp cells a maximum cytotoxic effect of 28.8% using the combination trastuzumab at 500 µg/mL plus pertuzumab at 2000 µg/mL (* *p* < 0.05) was achieved. (**D**) Similarly to the feline cell lines, in the human SKBR-3 cell line synergistic antiproliferative effects were obtained, revealing all the combinations valuable. The experiments were performed in triplicates and repeated three separate times.

**Figure 5 cancers-13-02007-f005:**
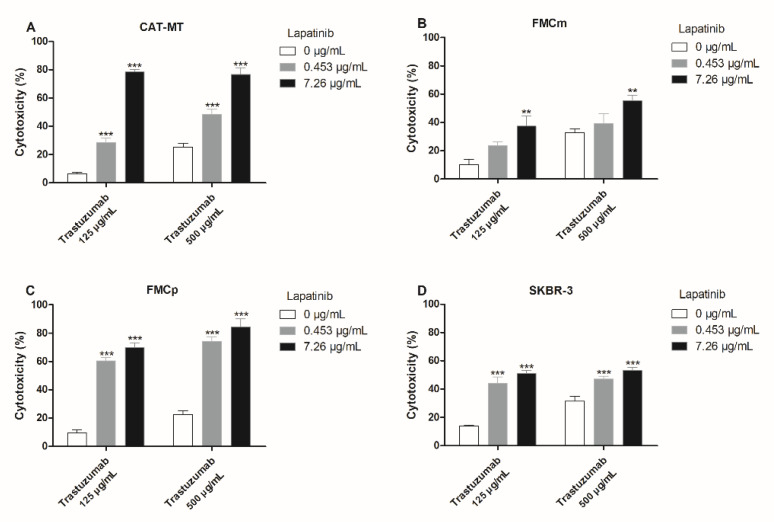
Combined treatments with trastuzumab plus lapatinib allowed to obtain a valuable synergistic antiproliferative effect in the carcinoma cell lines. (**A**) CAT-MT and (**C**) FMCp cells presented a significant increase in the cytotoxicity for all the concentrations tested (*** *p* < 0.001). (**B**) FMCm cells presented the highest increase in the cytotoxic effect by the combination trastuzumab at 125 µg/mL plus lapatinib at 7.26 µg/mL, being possible to achieve 37.4% of cytotoxicity (** *p* < 0.01). (**D**) In the human SKBR-3 cell line, all the combinations presented a synergistic antiproliferative effect (*** *p* < 0.001). The experiments were performed in triplicates and repeated three separated times.

**Figure 6 cancers-13-02007-f006:**
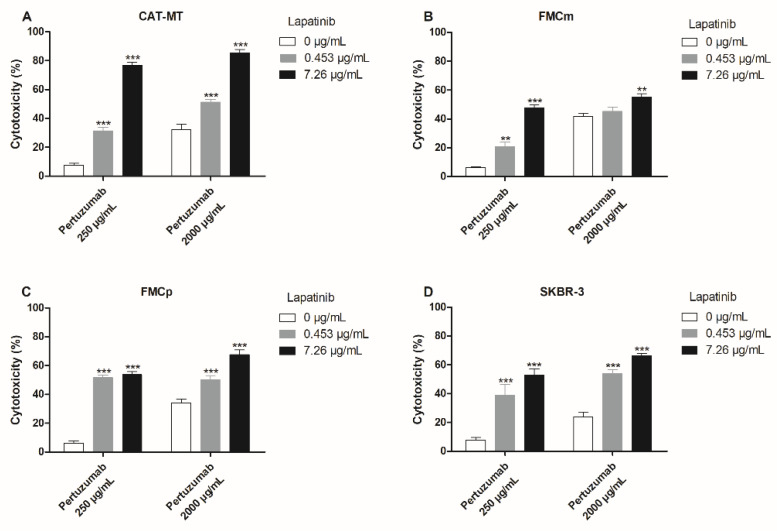
The combination of pertuzumab with lapatinib showed an improved antiproliferative effect in all the carcinoma cell lines. (**A**) CAT-MT cells presented synergistic antiproliferative effects for all the conjugations tested, the highest cytotoxic response occurring by the use of pertuzumab at 250 µg/mL plus lapatinib at 7.26 µg/mL, with 76.8% of cytotoxicity (*** *p* < 0.001). (**B**) The FMCm cells achieved the highest synergist effect, by the use of pertuzumab at 250 µg/mL plus lapatinib at 7.26 µg/mL, with 47.8% of cytotoxicity (*** *p* < 0.001), while the combination of pertuzumab at 2000 µg/mL plus lapatinib at 7.26 µg/mL presented a smaller cytotoxic effect (** *p* < 0.01). (**C**) The combination of pertuzumab at 250 µg/mL plus lapatinib at 7.26 µg/mL, in the FMCp cells, allowed to obtain 53.7% of cytotoxicity (*** *p* < 0.001). (**D**) As control, the human SKBR-3 cell line presented a synergistic effect for all the conjugations tested (*** *p* < 0.001). The experiments were performed in triplicates and repeated three separated times.

**Figure 7 cancers-13-02007-f007:**
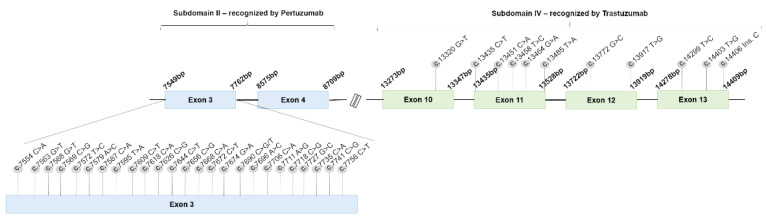
Mutations identified in the feline *her2* ECD, subdomains II and IV were not reported as induce resistance to therapy. All the mutations found occur in a low frequency (n = 1), being more common in exon 3, subdomain II (69.4%). Any of the described mutations were reported in breast cancer patients as induce therapeutic resistance.

**Table 1 cancers-13-02007-t001:** Concentrations of trastuzumab (µg/mL), pertuzumab (µg/mL) and T-DM1 (µg/mL) used in the cytotoxicity assays. Cells were exposed to drug for 72 h and the antiproliferative effects were evaluated.

Drug Concentrations for the Cytotoxicity Assays (µg/mL)		
Trastuzumab	Pertuzumab	T-DM1
25	50	5
50	100	12.5
125	250	25
250	500	50
500	1000	100
1000	2000	200
2000	5000	500
5000	10,000	1000
10,000		

**Table 2 cancers-13-02007-t002:** Concentrations of trastuzumab (µg/mL), pertuzumab (µg/mL) and lapatinib (µg/mL) used in the combined treatments. Cells were exposed to drug for 72 h before cytotoxicity evaluation.

Drug Concentrations for the Combined Treatments (µg/mL)		
Trastuzumab	Pertuzumab	Lapatinib
125	250	0.453
500	2000	7.26

**Table 3 cancers-13-02007-t003:** HER2 immunocytochemistry scoring criteria.

Score	Interpretation
0	No staining
1+	Weak, incomplete membrane staining
2+	Complete membrane staining, with obvious circumferential distribution in at least 10% of cells, that has either no uniform or is weak in intensity
3+	Uniform and intense membrane staining, at a minimum of 10% of the tumor cells

**Table 4 cancers-13-02007-t004:** Clinicopathological features of female cats with mammary carcinomas enrolled in this study (n = 40).

Breed	Number (%)	Age	Number (%)
Indeterminate	33 (82.5%)	<8 years old	3 (7.5%)
Siamese	4 (10%)	≥8 years old	37 (92.5%)
Persian	2 (5%)	**Tumor Size**	
Norwegian Forest	1 (2.5%)	<2 cm	9 (22.5%)
**Spayed (One Unknown)**		2–3 cm	19 (47.5%)
Yes	19 (47.5%)	>3 cm	12 (30%)
No	20 (50%)	**HP * Classification**	
**Contraceptives (Seven Unknown)**		Tubulopapillary carcinoma	8 (20%)
Yes	23 (57.5%)	Solid carcinoma	9 (22.5%)
No	10 (25%)	Cribiform carcinoma	5 (12.5%)
**Treatment**		Mucinous carcinoma	5 (12.5%)
Mastectomy	36 (90%)	Tubular carcinoma	11 (27.5%)
Mastectomy+Chemo	4 (10%)	Papillary-cystic carcinoma	2 (5%)
**Multiple Tumors**		**HP * Malignancy Grade**	
Yes	31 (77.5%)	I	2 (5%)
No	9 (22.5%)	II	5 (12.5%)
**Regional Lymph Node Status (Two Unknown)**		III	33 (82.5%)
Positive	14 (35%)	**Tumor Necrosis**	
Negative	24 (60%)	Yes	29 (72.5%)
**Stage (TNM Classification)**		No	11 (27.5%)
I	9 (22.5%)	**Lymphatic Invasion**	
II	7 (17.5%)	Yes	5 (12.5%)
III	21 (52.5%)	No	35 (87.5%)
IV	3 (7.5%)	**Lymphocytic Infiltration**	
**Mammary Location**		Yes	27 (67.5%)
M1	11 (27.5%)	No	13 (32.5%)
M2	8 (20%)	**Tumor Ulceration**	
M3	14 (35%)	Yes	3 (7.5%)
M4	11 (27.5%)	No	37 (92.5%)
**fHER2 Status**		**Ki67 Index**	
Positive	12 (30%)	Low (<14%)	30 (75%)
Negative	28 (70%)	High (≥14%)	10 (25%)
**ER Status**		**PR Status**	
Positive	12 (30%)	Positive	20 (50%)
Negative	28 (70%)	Negative	20 (50%)
**Tumor Molecular Subtype**			
Luminal A	3 (7.5%)		
Luminal B	18 (45%)		
Luminal B/HER2-positive	8 (20%)		
HER2-positive	4 (10%)		
Triple-negative	7 (17.5%)		

* HP—histopathological; TNM—tumor, node, metastasis; ER—estrogen receptor; PR—progesterone receptor.

**Table 5 cancers-13-02007-t005:** Primers for genomic DNA amplification and sequencing of exons 3, 4 and 10–13 of the feline *her2* ECD.

Exons	Forward (5′–3′)	Reverse (5′–3′)
3	GGCGCTTGCTCATAGTTCAC	ATCAAACTGTGCAGGCTCGT
4	GAGGCCTGCTCCCCTCTAAA	AAGAGGGAATGGGTAGCGTT
10–11	GGGCTTGGGCTTTGAAACTC	TGAAGGGTCAGCGAGTAAGC
12–13(1st pair)	TGGGAGTTTTCGGAGTGTGC	AAGCCTGACAGAAGGGATGG
12–13 (2nd pair)	GTGCTTACTCGCTGACCCTTCA	ACCCCTGCAATACTCGGCATTC

## Data Availability

The datasets used and analyzed in the current study are available from the corresponding author in response to reasonable requests.

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
