# Peer review of "HER2-Targeted Immunotherapy and Combined Protocols Showed Promising Antiproliferative Effects in Feline Mammary Carcinoma Cell-Based Models"

_cancers, 2021, doi:10.3390/cancers13092007_

Round 1
Reviewer 1 Report
This study is novel and gives a good insight into non-human breast cancers.
Introduction:
- The introduction is very clear. I suggest changing references to "women breast cancer" to "human breast cancer".
- Pertuzumab is never given as a single agent in humans. It would be better to lead with trastuzumab than pertuzumab.
- As trastuzumab, pertuzumab, and T-DM1 are humanised antibodies does this cause an immune reaction in cats?
Methods:
- The methods section is very clear and would allow for experiments to be reproduced.
- The k in SKBR3 should be capitalised.
- Cytotoxicity assays must be in biological triplicate rather than duplicate. This needs to be completed in order to carry out statistical analysis, as standard deviation of duplicate experiments is not valid.
- How many independent replicates were carried out for flow cytometry experiments?
Results:
- Figures are presented clearly. The addition of the cell line above each graph in figures 4, 5, and 6 would help though.
- T-DM1 cytotoxicity is low compared to other studies of SKBR3. Has this cell line been authenticated?
- Authors should carry out cytotoxicity analysis of an irreversible HER2-targeted TKI (i.e. neratinib, afatinib, pyrotinib). A TKI would be easier to administer to a cat and, importantly, would tie in better with their analysis of HER2 mutations in the tumour samples. Antibody therapies and lapatinib have minimum efficacy against HER2-mutant cancers. Cytotoxicity experiments with one these TKIs would connect the in vitro experiments and tumour sample analysis more wholly.
- Was HER2 amplification in the tumour samples examined?
Author Response
Reviewer 1:
This study is novel and gives a good insight into non-human breast cancers.
Dear reviewer, thank you so much for your positive comments about our work and your suggestions/corrections in order to improve the final quality of our manuscript. We hope that after the correction, you consider the manuscript suitable for publication.
Introduction
“I suggest changing references to "women breast cancer" to "human breast cancer".
Thank you for this remark. As suggested, the reference to “women breast cancer” was replaced by “human breast cancer” along with the manuscript.
“Pertuzumab is never given as a single agent in humans. It would be better to lead with trastuzumab than pertuzumab.”
Dear reviewer, we decide to start with pertuzumab to have a flow line along the text, since this mAb links to the ECD, subdomain II in the HER2 protein, but you are completely right, pertuzumab in humans is never used as a single agent and is better for the quality of the manuscript to start with trastuzumab. The Introduction section was changed, as you request (lines 59 to 75).
“As trastuzumab, pertuzumab, and T-DM1 are humanised antibodies does this cause an immune reaction in cats?”
Dear reviewer, thank you for this comment. Since we are working with cell line models, we used humanized antibodies with no concern about the immune reaction. However, for in vivo trials it will be important to have proper mAbs, adapted to cat (also called felinized, i.e. by grafting the complementary determinant region of the Fc Fragment) to prevent a possible immune reaction. As we mentioned in the Discussion section, there are few studies on the use of mAbs in cats and only one felinized antibody was reported for therapeutic proposes. A more detailed description about this issue was added to the Discussion section (lines 464 to 466), as well as proper references. Furthermore, a highlight about this concern was also added to the Conclusion section (lines 587 to 588).
Materials and Methods
“The k in SKBR3 should be capitalised.”
Dear reviewer, thank you for this correction, which was made along the manuscript.
“Cytotoxicity assays must be in biological triplicate rather than duplicate. This needs to be completed in order to carry out statistical analysis, as standard deviation of duplicate experiments is not valid.”
Dear reviewer, thank you so much for this highlight. Indeed, we had performed three independent assays (on different days), with triplicates for each data point. This information was correct in line 152, and in Figures 1, 4, 5 and 6.
“How many independent replicates were carried out for flow cytometry experiments?”
For the flow cytometry assay and, after the optimization of the technique, three independent experiments were performed (this information was added in the Materials and Methods section, line 199). The dot plot image is a representative image of the cytometry results.
Results
“The addition of the cell line above each graph in figures 4, 5, and 6 would help though.”
As you suggested, the cell line name was added above each graph, in Figures 4, 5 and 6. Thank you for this comment that helps to read the figure.
“T-DM1 cytotoxicity is low compared to other studies of SKBR3. Has this cell line been authenticated?”
Dear reviewer, this cell line was purchased from ATCC (Rockville, Maryland, USA) and gently provided by Prof. Luís Costa (Institute of Molecular Medicine, Lisbon, Portugal). This information was added to the Materials and methods section (lines 125 to 130).
“Authors should carry out cytotoxicity analysis of an irreversible HER2-targeted TKI (i.e. neratinib, afatinib, pyrotinib). A TKI would be easier to administer to a cat and, importantly, would tie in better with their analysis of HER2 mutations in the tumour samples. Antibody therapies and lapatinib have minimum efficacy against HER2-mutant cancers. Cytotoxicity experiments with one these TKIs would connect the in vitro experiments and tumour sample analysis more wholly.”
Dear reviewer, we understand your suggestion and thank you for this comment that emphasizes the importance of this kind of experiment. In fact, despite the secondary effects of the TKIs, in a practical point of view, they are easier to administrate and cheaper than mAbs. As suggested, the cytotoxicity assays using a reversible TKI (lapatinib) and an irreversible TKI (neratinib), were already performed and published by us (https://doi.org/10.3390/pharmaceutics13030346). Interestingly. by the use of these compounds, particularly neratinib, very promising results were obtained in all FMC cell lines, including in the HER2-negative one. Furthermore, similarly to the results of the her2 ECD sequence analysis, no mutations were found in the her2 TK domain, suggesting no cases of therapy resistance. A brief explanation considering the importance of the TKIs and a proper reference was added to the Discussion section (lines 466 to 471).
“Was HER2 amplification in the tumour samples examined?”
Dear reviewer, thank you so much for this remark. The HER2 status in these FMC clinical samples was previously evaluated and published by our group (e.g. serum HER2 levels and HER2 tissue expression - https://doi.org/10.18632/oncotarget.7551; her2 gene amplification - https://doi.org/10.1017/S1431927613001529). Interestingly, these studies revealed that, in FMC, the increased expression of HER2 is not associated with her2 gene amplification. A brief description and a proper reference were added to the Materials and Methods section (lines 225 to 227).
Reviewer 2 Report
To the authors
The present manuscript entitled HER2 targeted immunotherapy and combined protocols showed promising antiproliferative effects in feline mammary carcinoma cell-based models clearly describes the in vitro effects of mABS alone or with lapatinib in feline cell lines derived from mammary carcinomas.
The experiments are well conducted and tha data shown are convincing and can really represent a starting point for further potential therapies in the treatment of feline mammary carcinomas.
Neverthless there are some critical points that need to be clarified in order to improve the scientific value of the present manuscript.
- Simple Summary
Lines 16-19 please change the phrase “ with cat …………………” where the absence of mutation ………….suggest ………………..”
- Introduction
I’s too long please slightly syntethize the part relative to human breast cancer and insert also data on AKT in feline mammary carcinomas ( Maniscalco et al. 2012 The VETH JOURNAL). These data need to be discussed also in relation to possible pathways HER2 dependent in feline cell lines and mTOR expression.
- Materials and methods
Line 110 specific references to feline mammary cell lines used need to be inserted and also the type of tumor origin of the cells.
In the in vitro cytotoxicity assays the authors did not indicate the number of cells treated in the experiments. Regarding this aspect it should also important insert for all cell lines analyzed the EC50 at dosage and time used in the assay.
- Results
The authors assessed that FMCp show a low level of HER2 but this cell line responds to mAb treatment going to apoptosis process and decrease the proliferative effects. How did the authors explain this in vitro behaviour? How did they explain this biological feature??
I suggest to insert in supplementary materials images about IHC (HER2-ER-PR-Ki67) performed on feline mammary tissues
DISCUSSION
Also the discussion is too long. Please synthetize and insert also data on AKT expression in feline mammary carcinomas in relation to data obtained by the authors (see comment in the introduction)
In consideration of the pathways mediated by HER2 did the authors perform evaluation by Western Blot specific for p-AKT and mTOR in treated cell if compared to control cell?? These data if inserted should improve the value of the manuscript and demonstrate the same mechanism known in SKBR3.
Tablese and images: OK
Reference
The ref 12 is wrong. The first name in R. De Maria. Please correct.
Author Response
Reviewer 2:
The present manuscript entitled HER2 targeted immunotherapy and combined protocols showed promising antiproliferative effects in feline mammary carcinoma cell-based models clearly describes the in vitro effects of mABS alone or with lapatinib in feline cell lines derived from mammary carcinomas.
The experiments are well conducted and tha data shown are convincing and can really represent a starting point for further potential therapies in the treatment of feline mammary carcinomas.
Dear reviewer, thank you so much for your review, comments, and time in order to improve the quality of our manuscript, and also for giving positive feedback on our work.
Simple Summary
“Lines 16-19 please change the phrase “with cat …………………” where the absence of mutation ………….suggest ………………..”
Thank you for this comment. In fact, in the used feline cell lines we found a small number of mutations, with none described in human breast cancer with therapy-resistance. As you suggested, the sentence was corrected (lines 18 to 19).
Introduction
“I’s too long please slightly synthetize the part relative to human breast cancer and insert also data on AKT in feline mammary carcinomas (Maniscalco et al. 2012 The VETH JOURNAL). These data need to be discussed also in relation to possible pathways HER2 dependent in feline cell lines and mTOR expression.”
Dear reviewer, thank you for this correction. All the Introduction section was revised in order to synthesize it. Furthermore, the reference you recommend was added and discussed considering the importance of the AKT and mTOR pathways in the feline mammary carcinoma, not only in the Introduction section (lines 44 to 46; and lines 59 to 62), but also in the Discussion section (line 477; lines 481 to 483; and line 486 to 487).
Materials and Methods
“Line 110 specific references to feline mammary cell lines used need to be inserted and also the type of tumor origin of the cells.”
As suggested, two additional references were added (lines 128). Furthermore, the type of tumor origin of the cells was added in the Supplementary materials, Table S1.
“In the in vitro cytotoxicity assays the authors did not indicate the number of cells treated in the experiments.”
Thank you for this important remark. As suggested, the number of cells seeded was added for each cell line and for each experiment (lines 145 to 146; and lines 196 to 197).
“Regarding this aspect it should also important insert for all cell lines analysed the EC50 at dosage and time used in the assay.”
The dosage used in each assay was added to Table 1, with these assays occurring in a period of 72 hours, as described in lines 148 to 149. As you suggest, an improvement of the cytotoxicity analysis was made, by calculating the EC50 values. This information was added to the Statistical analysis – Material and Methods section (lines 274 to 275). Furthermore, the EC50 values for each cell lines, were added to the Results section (lines 284 to 295), as well as an improved Figure 1.
Results
“The authors assessed that FMCp show a low level of HER2 but this cell line responds to mAb treatment going to apoptosis process and decrease the proliferative effects. How did the authors explain this in vitro behaviour? How did they explain this biological feature??”
Dear reviewer, thank you for these questions. In fact, we obtained some interesting results by testing this HER2-negative cell line (FMCp). Revising the literature, some studies suggest that the HER2-negative tumors are not completely negative and could present activated HER2 forms or circular HER2 RNA that encodes for HER2-103, which putatively occurs in FMCp cell line. These questions were answered in the Discussion section and some references were added (lines 486 to 490).
“I suggest to insert in supplementary materials images about IHC (HER2-ER-PR-Ki67) performed on feline mammary tissues”
Thank you for this suggestion. The immunocytochemistry images (ER, PR, Ck5/6 and Ki-67) for all the cell lines were inserted in the Supplementary materials, as Figure S1. Concerning the HER2 expression, it was represented in Figure 2 of the Results section.
Discussion
“Also the discussion is too long. Please synthetize and insert also data on AKT expression in feline mammary carcinomas in relation to data obtained by the authors (see comment in the introduction).”
Dear reviewer, all the discussion section was revised in order to summarize it. Furthermore, a proper explanation on the trastuzumab effect in the FMCp cell line was added, as you suggested (lines 486 to 490), as well as, a brief discussion about the AKT expression and function in the FMC (line 477; lines 481 to 483; and line 486 to 487).
“In consideration of the pathways mediated by HER2 did the authors perform evaluation by Western Blot specific for p-AKT and mTOR in treated cell if compared to control cell?? These data if inserted should improve the value of the manuscript and demonstrate the same mechanism known in SKBR3.”
Dear reviewer, in fact, this kind of data will improve the manuscript, but unfortunately, we don’t have the chance to get additional samples of mAbs targeting HER2, from medical hospitals. Additionally, the research project responsible for the financial support of this work has finished, making it impossible to buy additional reagents. Thank you for your understanding.
References
“The ref 12 is wrong. The first name in R. De Maria. Please correct.”
Dear reviewer, reference 12 was corrected.
Round 2
Reviewer 1 Report
The authors have addressed the majority of issues raised in the first review and with some minor revisions should be accepted for publication.
The difference in T-DM1 response in SKBR3 compared to previous literature must be acknowledged in the results or discussion. See DOI: 10.1158/1078-0432.CCR-17-0696 for comparison.
Error bars need to be included for bar charts in Figure 3.
Please reference your previous study of lapatinib/neratinib in these cell lines in Section 3.3 or in the discussion.
Author Response
Reviewer 1:
Dear reviewer, thank you for your remarks in order to improve the final quality of our manuscript. We hope that after this second round of corrections you can consider this manuscript suitable for publication.
“The difference in T-DM1 response in SKBR3 compared to previous literature must be acknowledged in the results or discussion. See DOI: 10.1158/1078-0432.CCR-17-0696 for comparison.”
The results obtained for SKBR-3 cell line exposure to T-DM1 were described and compared to the literature in the Discussion section, and additional references were added (lines 480 to 484). Thank you for the reference you gave as an example, which was taken into account.
“Error bars need to be included for bar charts in Figure 3.”
Error bars were included in the Figure 3, as suggested. Furthermore the standard deviation was also included in the flow cytometry results (lines 293 to 314).
“Please reference your previous study of lapatinib/neratinib in these cell lines in Section 3.3 or in the discussion.”
The statement included in the Discussion section was improved considering the results obtained by the use of lapatinib and neratinib in the feline cell lines (lines 436 to 440), as well as the importance of combined therapeutic protocols, as the use of TKI plus mAbs (lines 514 to 519), as requested. Finally, references supporting the introduced statements were also added (line 436 and 517), including our last published study.